# Evidence from UK Research Ethics Committee members on what makes a good research ethics review, and what can be improved

Mark Sidaway[1], Clive Collett[1], Simon Erik Kolstoe[2]*

1 Health Research Authority, Stratford, London, United Kingdom, 2 School of Health and Care Professions, University of Portsmouth, Portsmouth, United Kingdom

* simon.kolstoe@port.ac.uk

## Abstract

The rapid development of vaccines and other innovative medical technologies in response to the COVID-19 pandemic required streamlined and efficient ethics and governance processes. In the UK the Health Research Authority (HRA) oversees and coordinates a number of the relevant research governance processes including the independent ethics review of research projects. The HRA was instrumental in facilitating the rapid review and approval of COVID-19 projects, and following the end of the pandemic, have been keen to integrate new ways of working into the UK Health Departments' Research Ethics Service. In January 2022 the HRA commissioned a public consultation that identified strong public support for alternative ethics review processes. Here we report feedback from 151 current research ethics committee members conducted at three annual training events, where we asked members to critically reflect on their ethics review activities, and to share new ideas or ways of working. The results showed a high regard for good quality discussion among members with diverse experience. Good chairing, organisation, feedback and the opportunity for reflection on ways of working were considered key. Areas for improvement included the consistency of information provided to committees by researchers, and better structuring of discussions by allowing signposting of the key issues that ethics committee members might need to consider.

## Introduction

Research is a progressive endeavour. Systems and processes designed to ensure research, and researcher, accountability must be agile and evolve in parallel with emerging methodology and technology. The review of research projects by Research Ethics Committees (RECs), also known as Institutional Review Boards (IRBs) in some countries, is one important element of ensuring research accountability [1]. There is an increasing literature that critically examines the activities of RECs by considering issues such as consistency [2], the overall reviewing remit of committees [3], and how they integrate with research governance activities [4]. This reflection on the role and function of committees was accelerated during the COVID-19 pandemic

**Data Availability Statement:** The raw data, grouped into categories, is included in the Supporting Information.

**Funding:** The work described here was conducted as normal duties for MS and CC as part of their employment with the Health Research Authority (HRA). SEK was seconded to the Health Research Authority to conduct this work, and was remunerated through the University of Portsmouth. As employees of the HRA this work was designed, and data collected, by the funder, however the analysis, decision to publish and preparation of the manuscript was conducted independently.

**Competing interests:** MS and CC work for the UK Health Research Authority. SEK was seconded to the HRA to conduct this research, and is also chair of the Cambridgeshire and Hertfordshire Research Ethics Committee. SEK is also a trustee of the charity UKRIO. This does not alter our adherence to PLOS ONE policies on sharing data and materials.

due to the intense pressure on all parts of the research system to produce medical and technological solutions in record time [5]. Previous work on research waste (where research activities do not lead to published results) have highlighted how lengthy ethics or approval processes can hinder productive research [6], and thus it has been important to record and reflect on the solutions developed during COVID-19 to see if they can be applied more widely.

The Health Research Authority (HRA) coordinates and standardises regulatory practice across the UK for all research falling under Clinical Trials Regulations (alongside the Medicines and Healthcare products Regulatory Agency), or more broadly involving the UK's National Health Service (NHS). Within England (there are separate REC arrangements for Scotland, Wales and Northern Ireland) the HRA's Research Ethics Review Service appoints and coordinates 60 to 70 RECs [7]. These are constituted by up to fifteen members composed of those with professional medical or medical research experience (designated "expert" members), and at least a third who are designated as "lay" members with no such direct professional experience. Four to six applications are reviewed at monthly meetings conducted online in accordance with policy and standard operating procedures provided by the HRA. An "*Ethics Review Form (ERF)*" is also provided with headings [8] that RECs are expected to consider when reviewing a project.

In order to develop their policies and further improve ways of working, the HRA regularly consults widely among the research community, ethics committees and society more generally. Following the 2020/21 SARs-CoV-2 pandemic, the HRA established the "*Think Ethics*" programme to consider whether the pandemic had highlighted any new policies and/or ways of working that might be needed, and if so to consult on possible changes [9]. This project was consistent with similar reflections by many others within the UK and international research environment following the pandemic experience [10–12]. Among the first activities conducted by the HRA was a public dialogue exercise in January 2022. This recorded four main findings [13]:

1. *that there is widespread support for alternative ethics review methods*

2. *the importance of diversity within RECs so as to promote inclusivity and diversity within research participation*

3. *a need for increased visibility of the activities overseen by the HRA including REC review*

4. *ongoing ethical monitoring of research beyond the current, initial, approvals process*

The findings from this public dialogue exercise suggested an appetite for changes to the ethics review process, but the opinion of public contributors do not represent the opinion of those experienced in the practicalities of conducting research ethics reviews, or indeed the research community itself. It was therefore important for the *Think Ethics* programme to consider the views of REC members alongside the public opinion.

As a public authority tasked with overseeing RECs, the HRA does regularly collect feedback from committee members through processes such as the "Shared Ethics Debate" [8], but for the purposes of the *Think Ethics* programme additional focus groups were held to explore the issues raised by the public dialogue exercise. These were organised as one of a number of activities held during annual face to face "*REC members' development days*". The development day format was new to the HRA and their committees in 2022, based on the pandemic forcing the majority of REC meetings online via video conferencing software. While this online reviewing arrangement has been found to work well for the ethics review of projects themselves (representing a major new way of working discovered due to the pandemic), the HRA was keen to act on feedback from REC members asking for the opportunity to meet other members face to

**Table 1. REC participants at each of the three member development days.**

|  | Birmingham | Reading | London | TOTAL |
|---|---|---|---|---|
| REC members | 31 | 42 | 49 | 122 |
| REC Officers (Chairs, Vice-Chairs & Alternate Vice-chairs) | 6 | 10 | 13 | 29 |

face at least once a year to discuss relevant topics. In 2022 the topics under discussion included updates from HRA staff, a presentation on how to support adults with capacity or communication difficulties in the consent process, a presentation on updates to clinical trials legislation, and a session entitled "*Think Ethics*" run by the authors, as a reflection on current ways of working, along with an opportunity to raise new ideas for the conduct of ethics reviews. Here we report our findings from these latter sessions.

## Materials and methods

### Ethics review

The work described here was conducted as part of ongoing service improvement by the Health Research Authority and carried out by the authors as part of their normal employment. It therefore did not require an ethics review in accordance with the UK's governance arrangements for Research Ethics Committee's policy (section 2.3.13 & 2.3.14) [14]. As all data was recorded anonymously specific consent was not required under data protection legislation, although all participants were verbally informed as to the purpose of the session, and the fact that data would be collected, analysed and published.

### Setting/Participants

Data was gathered during three one hour sessions held at REC members development days in Birmingham (8th September 2022, 37 REC members present), Reading (20th October 2022, 52 REC members present) and London (24th November 2022, 62 REC members present). Attendance data, including which REC regions were represented, was gathered by the HRA and shown in Table 1.

### Design/Procedure

Our design can broadly be considered a consultation exercise, with participants split into groups of six to ten REC members (including a mix of committees) and, following a brief ten minute introduction to the "Think Ethics" programme, asked to discuss the following five questions under the heading "*Reflecting on our ethics reviews*". The questions were chosen following discussion among the authors with input from the HRA training team, and were designed to elicit discussion and reflection within the groups (so similar to a semi-structured focus group):

1. How do we get to our ethics opinion?

2. What are the most important questions to ask?

3. What is the purpose of the ethics review form (described below)?

4. Are there different ways of reaching this opinion?

5. How do we satisfy ourselves that we have done a good job?

Discussions were allowed to flow freely for approximately 30 minutes with a note taker in each group (most often a HRA member of staff) summarising the discussions in the form of

bullet points. A twenty minute feedback exercise was then conducted allowing each group to state their main thoughts and conclusions to the whole room. Summary notes from this latter session were also recorded by one of the authors, again as bullet pointed notes. As each of the three sessions were conducted in large conference rooms it was not possible to make audio recordings. All quotations referenced in the results are therefore summarised notes rather than a verbatim record of original statements (see limitations section).

## Analysis

Following the REC members development events, the bullet pointed notes from each group, and notes from the final feedback sessions, were combined under the five questions as above (note takers were asked to format their notes under these headings). In order to analyse the data, and mindful of the spectrum of methodologies that can be used to analyse such data [15–17], a content analysis method similar to recent work considering improvements in REC/IRB reviews [18] was chosen which included both a quantitative element (number of statements coded to each category) and subsequent reflexive consideration of both the identified categories and subsequent statements therein. To do this two separate researchers initially read through the data individually, creating codes/categories assisted by the NVivo [19] software. Consensus codes/categories were then agreed in discussion between the two researchers, along with the statements that were included in each category. The coded data can be found in the S1 Data. It must be reiterated that statements referred to in the results are not verbatim, but rather represent summarised bullet points made by the note takers.

## Results

Tables 1 and 2 describes the participants and their REC regions, demonstrating a good geographic spread across England. Note East of England, North West and Social Care REC numbers were combined due to low numbers and to preserve anonymity.

### Question 1: How do we get to our ethics opinion?

There were 75 statements linked to this question (by note takers) that were subsequently coded to 14 categories or sub-categories (see Table 3). Six references were coded to more than one category giving 81 references overall. By far the most common category was the role of discussion of research proposals during the REC meeting, with six comments referring to the role of the "lead reviewer" (a committee member who takes responsibility for introducing a study) and a number of further comments on the importance of the REC chair in leading the discussion. There were mixed views as to whether face to face or online formats assisted the discussion, however, all the comments under this category viewed having a discussion as a positive aspect of the review process. The importance of listening to other members, and the observation that consensus was commonly reached, was also notable. Use of the online ethics review

**Table 2. Representation from different REC regions.**

| REC region | Number of Members Present |
|---|---|
| East Midlands | 16 |
| East of England, North West and Social Care REC | 10 |
| London | 51 |
| South Central | 40 |
| South West | 13 |
| West Midlands | 21 |

**Table 3. Categories emerging in answer to the question 1: How do we get to our ethics opinion?.**

| Name | References |
|---|---|
| Consistency | 5 |
| Ethics Review Form | 10 |
| Focus on Participant Information | 7 |
| Focus on Recruitment | 1 |
| Focus on Research Design | 6 |
| Focus on theory | 6 |
| Ethics Domains | 1 |
| Risk/Benefit | 4 |
| Importance of Discussion | 23 |
| Lead Reviewer | 6 |
| Role of REC chair | 3 |
| Zoom vs F2F | 5 |
| Patient & Participant Views | 1 |
| Proportionate Approach | 3 |

form (ERF), which suggests ten domains to be considered when conducting an ethics review, was the second most common category, again with all comments positive (see results for question 3 for a more detailed description of the ERF). Members also reported that they tended to focus on the participant information, research design and ethical theory (in terms of "Principlism" or "Deliberative Theory"), while practicalities such as trying to be consistent and proportional were also made.

## Question 2: What are the most important questions to ask?

There were 63 responses to this question, and once coded (Table 4) almost a third or references related to questions about risk and safety. The second most common category were questions directly relating to the design and justification of the research including whether it was articulated in a way that REC members could understand. There was also a number of acknowledgements, and apparent confusion, as to the difference between an "ethics" review and a "scientific" review, and the role of other regulators such as the Medicines and Healthcare products Regulatory Agency (MHRA) whose role is specifically to look at safety and methodology e.g.

> "*Is poor research design a material ethical issue?*"

**Table 4. Categories emerging in answer to the question 2: What are the most important questions to ask?.**

| Name | References |
|---|---|
| Consistency & Clarity of Information | 6 |
| Consistency of Review Process | 7 |
| Different Perspectives | 3 |
| Role of Chair | 1 |
| Participant Information & Consent | 9 |
| Participants Perspective | 5 |
| Recruitment Strategy | 1 |
| Research Design & Justification | 11 |
| Researcher's attitude & experience | 5 |
| Risk & Safety | 19 |

*"how much science to scrutinise, how much is the MHRA concern?"*

*"does the study answer the research question—fundamental. Doing something that isn't going to produce a valid good result is pointless"*

Checking the clarity of the participant facing information was also a distinct category, linked with the idea of trying to look at the information from the participant's perspective, therefore ensuring it is clear and understandable to the "lay" person. Two categories relating to consistency were also raised, firstly the consistency of the information presented to the REC by the researchers, and secondly consistency of the questions that the committee considered, with a concern that committees need to stay focussed on fundamental issues, rather than being distracted by minor issues (such as phraseology). There were also a few comments on the importance of having a diverse committee and the value that could bring:

*"The broad range of views and opinions on the REC, the group variety makes for good ethical decisions. You look at studies from many different angles. For example, a lay member could ask an obvious question that has not been asked or assumed."*

## Question 3: What is the purpose of the ethics review form (ERF)?

The ethics review form (ERF), referred to by some as the "Lead Reviewer's Form", was originally a paper document appended to study documents by the HRA, but sent only to one or two REC members so that they could act as lead reviewers. It contained ten areas that the REC should consider when reviewing studies. These areas were developed over a number of years by the HRA (and its predecessor organisations) and were summarised by the National Research Ethics Advisors Panel, as described in reference [8]. As documents are now provided entirely online via the "HRA Assessment Review Portal" (HARP) system, the ten headings from the original ethics review form have been added as a tab associated with each study, with all members now able to access and add comments online (see Fig 1). The collated comments can be downloaded and used during the REC meeting, most often led by the lead reviewer or chair working their way through the headings.

There were 55 references to the ERF (see Table 5) with the most common being that the form assisted in the review process by helping to structure discussions or subsequent minutes, and perhaps aid consistency:

*"a good way to structure the conversation. It's important to raise the issues. Consistency."*

There were eight references in total discussing whether the headings were either suitable (3 references) or not (5 references). The second most common category was to the form being very useful when preparing for meeting (11 references) along with being helpful for the HRA staff who are managing the meeting:

*"This is used to communicate with other members before the meeting and stimulate discussion."*

There were eight references noting that not all members use it, and a further seven commenting that its structure does not align particularly well with application forms or study documentation provided by the researchers.

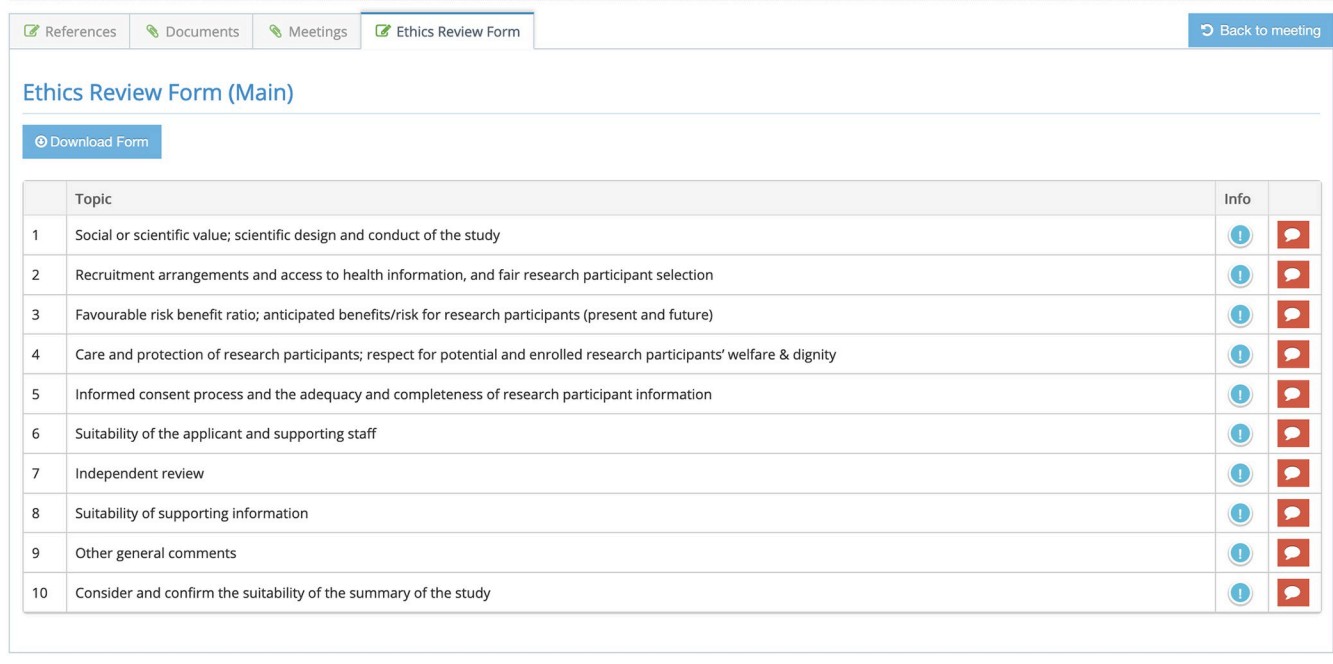

**Fig 1. The ethics review form tab on HARP.**

### Question 4: Are there different ways of reaching this opinion?

There were 22 references under this question (Table 6) with the most common being to note that different REC members approached reviewing studies in different ways such as starting with the participant information sheet (PIS) rather than the ethics application form or protocol:

"*Some people start with the PIS (as if they were a participant), rather than the protocol.*"

Another seven references related to the way discussions were structured or flowed noting that the lead or second reviewers were important for leading discussions, and a further four references commenting that the committee chair is quite influential:

"*Different chairs, different methods. Chairs ask, tell, lead reviewer does this, group discussion and occasionally voting if it comes to it.*"

**Table 5. Categories emerging in answer to the question 3: What is the purpose of the ethics review form?.**

| Name | References |
|---|---|
| Conflict between documents | 7 |
| Helps new members | 1 |
| New section | 2 |
| Not everyone uses it | 8 |
| Prepare for meeting | 11 |
| Structures or Shortens review | 18 |
| Right topics | 3 |
| Wrong topics | 5 |

**Table 6. Categories emerging in answer to the question 4: Are there different ways of reaching this opinion?.**

| Name | References |
|---|---|
| Difference between members | 3 |
| Co-opting members | 1 |
| Different backgrounds | 1 |
| Different approach to documents | 7 |
| Structure of discussion | 7 |
| Role of Chair | 4 |

These categories seem related to the five references commenting that members' background and experience affects how they approach the studies and discussion.

## Question 5: How do we satisfy ourselves that we have done a good job?

This question generated the most feedback with 44 distinct comments coded as 47 references (Table 7). The most common category (15 references) was the importance of receiving feedback from researchers, the HRA or other committee members, with a further five references commenting that committees try to reflect or evaluate on their own workings, including how the conversations went with the researchers:

> "*At the end of the REC meeting, do a wash up session—did we do a good job or do we feel weary?*"

And:

> "*Was it a good interaction with the applicant, was there good dialogue?*"

There also seemed to be confidence in the system/processes of review such as the varied membership of committees (4 references), and the management of the committees by the HRA including time management and the efficiency of the process, although the absence of breaks was noted.

**Table 7. Categories emerging in answer to the question 5: How do we satisfy ourselves that we have done a good job?.**

| Name | References |
|---|---|
| Committee reflection | 5 |
| Comparison with other RECs | 3 |
| Confidence from talking to researchers | 3 |
| Feedback from stakeholders | 15 |
| Reactive/evolving process | 1 |
| Reassurance from contribution of other members | 4 |
| Seeing changes proposed | 1 |
| Seeing changes to protocols | 2 |
| Seeing outcomes of research | 5 |
| Site inspection | 1 |
| Timely & Efficient process | 4 |
| Very few research disasters | 3 |

There was also a feeling of reassurance that came from seeing or hearing about the results of high quality, useful, research, and the fact that there are very few problems reported with studies that have been reviewed by the RECs:

"*The system is not broken. The RECs have reviewed 1000s of studies. There have been few if any disasters post REC review. There is always scope to reflect and improve the reviews, but the system is not broken and does not need to be replaced.*"

## Discussion

This consultation exercise represented a valuable opportunity to hear directly from REC members as to how they perceive their work, and what they think could be improved. While stakeholders such as researchers are often keen to see REC review streamlined as much as possible [20], and the earlier *Think Ethics* public consultation indicated public support for reviewing ways of working, in general the REC members who contributed to this exercise were satisfied that their committees were doing a good job of conducting ethics reviews. The evidence described here is helpful to draw out why REC members think this to be the case, but also to guide future evidence-based improvement. While specifically relevant to HRA coordinated RECs in England, these findings are also broadly applicable within other contexts such as University RECs that often follow the extensive guidance and principles established by the HRA and promulgated through its website. Indeed within an international context, it is worth noting that the HRA network of RECs is the most extensive in the world, and thus more should be done to share their findings more widely in support of smaller, or less well coordinated, REC/IRB systems.

Perhaps the first generalisable finding from this work was that good quality and well-structured discussions within committees are key to high quality ethics reviews, and while there were plenty of comments relating to why high-quality discussions were not always achieved, the need to both promote (perhaps through training) and support (perhaps through improved ways of working) such discussions was clearly highlighted. Here the UK's advantage of having a national system can clearly be seen as such training can be incorporated into the existing programme of REC member support as implemented by the HRA. Our results indicate that the content for such training could focus specifically on the role of either chairs or lead reviewers in leading the discussions, the way that expertise (either scientific or lay) is able to feed into such discussions, and how discussions are structured. This latter point fits closely with comments relating to the Ethics Review Form (ERF), and also answers to question three as to how the ERF is specifically used. While the headings were broadly seen as helpful (covering broad issues like scientific justification, consent process, dissemination plans etc, see Fig 1), the format of the form was viewed as problematic by some, and likewise the inconsistent approach between committees, and even members on the same committee, was highlighted. More work could therefore be done to improve the way that the ERF is used.

A second major theme coming out of this work relates to how information is presented to committees, and inconsistencies between documents. RECs are well known for combing through documents in detail, particularly participant information sheets, to pick up inconsistencies some of which are not always particularly relevant to the ethical acceptability, or otherwise, of the proposed research [8]. However, inconsistencies are very relevant if they occur within the protocol, or between the protocol and participant information/consent documents, as they might lead to confusion both within research teams as they conduct the work, and for participants who could potentially be misled as to what they are being asked to do (and critically consent to). Discovering and highlighting these ethically relevant inconsistencies is an

important role for the REC, but not helped if the review documentation is provided in a piecemeal fashion using poorly structured templates. Here the discrepancy in how information is presented between protocols, the ethics application form and particularly the ERF form was highlighted. Given that all three documents aim to present and/or capture similar content, one improvement could be the harmonisation of the templates to ensure information such as study justification, study methodology, recruitment processes, risks to participants etc. are presented in roughly the same order. While the complexity and diversity of study designs must also be acknowledged, along with inconsistencies in the way that individual researchers may approach and write about their own specific protocols, more could be done from an administrative perspective to force as much consistency as possible in terms of the information is presented to REC members.

A third important issue emerging from this work was how committee members viewed themselves and their role. The UK wide Governance arrangements for research ethics committees (GAfREC) [14] define the membership of committees to allow for a sufficiently broad range of experience and expertise so that the rationale, aims, objectives and design of the research proposals can be effectively reconciled with the dignity, rights, safety and well-being of the people who are likely to take part. However, as not all members can attend every meeting, the actual composition of the REC at any single meeting may vary, thus subtly changing the balance of views brought to bear on any application under review. Likewise, previous work has shown that discussions can stray into overly scientific or medical areas rather than focussing on key ethical issues [8]. Here answers to question 4 (*Are there different ways of reaching this opinion*?) were particularly informative as they demonstrated that committee members do have an appreciation of the different perspectives that members from different backgrounds bring, and that they do often rely on each other for picking up different aspects of the review. Diversity is therefore important within committees. However, it is unclear as to how this diversity in approaches (and topics) considered by individual members may relate to broader issues around cultural and social diversity. For example, REC diversity data collected by the HRA [21] clearly indicate a predominantly female (62%) and older (70% above 55) demographic with lower representation from African, Caribbean, and Arab ethnicities in particular. Likewise GAfREC [14] states that a "REC should contain a mixture of people who reflect the currency of public opinion ('lay' members), as well as people who have relevant formal qualifications or professional experience that can help the REC understand particular aspects of research proposals ('expert' members)". However, while the breadth of experience is clearly important, all REC members must also be able to read and understand at least basic scientific ideas and terminology so that they can take part in the discourse that is central to the REC review process. As a consequence, the role of Patient and Public Involvement and Engagement (PPIE) has increasingly been viewed as another central, and complementary, aspect of research review [22] alongside the parallel reviews provided by other more specialist governance/regulatory reviews (such as provided by the MHRA) and scientific peer review. It was therefore very reassuring to find that the importance of the PPIE was indeed picked up as a category under question 2 (*What are the most important questions to ask*?) where REC members were particularly reassured when they saw good PPIE. This observation highlights the importance of having a better understanding of what it means for a REC to 'reflect the diversity of the adult population of society' and 'current ethical norms in society as well as their own ethical judgement' (as required by GAfREC) and how this facilitates robust ethics review, and the diversity required across the research process as a whole (incorporating REC, PPIE and other reviews).

The answers given to question 5 (*How do we satisfy ourselves that we have done a good job*?), were also very interesting. Direct feedback from stakeholders such as researchers (often

through direct interaction with the committee as they attend virtual meetings), and from the HRA itself, seemed to provide a straightforward and obvious means of validating a committee's activities. It should also be noted that the HRA regularly ask for, and pass on, any further written feedback from researchers. However, perhaps more interesting was the way committee members were able to reassure each other, and also the way that members enjoyed co-opting on to other RECs and so being reassured that their REC was operating in broadly the same way as others. Indeed one interesting consequence of moving to an almost exclusively online meeting structure due to the COVID-19 pandemic, has been the break between geographic location and the REC that members are able to contribute to. As can be seen in Table 2, RECs are predominantly still based in geographic region, and attendance at the REC training days used for this project also indicated that members from specific committees are also still predominantly located close by. Over time as membership changes and travel is no longer a relevant concern, it might be anticipated that a more distinct break between geography and committee membership might be seen perhaps eventually even necessitating the renaming of committees, although the needs of researchers (and their familiarity with certain committees) would also need to be considered should this sort of decision ever be made.

## Limitations

As with all qualitative research the results are descriptive of the views of the people in the room, and in this case as all participants were active, and volunteer, REC members, it is likely that they provided a more positive view of REC performance compared to if this exercise had been conducted with other groups. Furthermore, multiple note takers were used to summarise the statements originally made by the REC members, meaning that all the data analysed here has been subject to interpretation by the note takers who were mostly administrative members of HRA staff. However, given the consistency between the notes that we received, it seemed that rather than adding anything to the original comments, the main problem that this introduced was more likely linked to missing contextual information for the sake of brevity. For instance noting single words like "risk" or "consistency" without the wider context of how these issues were actually discussed at the time. Given this potential ambiguity, we have been careful in our analysis to only group comments under the specific question headings as given to us by the note takers, despite the fact that some comments could potentially have applied to multiple questions. However, it was reassuring to see similar categories picked up across the three separate events, and we have tried to consider overall themes emerging from all five questions in our discussion.

## Conclusions

It is worth noting that although there is a considerable literature considering how RECs review studies [23, 24], this literature is seldom explicitly read or referenced by the members of committees who often "learn on the job". As a consequence, the observations recorded in this study represent a useful summary of how existing members currently perceive their role, and ways that the processes linked to ethics review could be improved from the perspective of those actually conducting the reviews. It is also important to note that in light of technology, policy and administrative processes always changing, and especially following the COVID-19 pandemic experience, this work provides timely guidance that can be used to guide future improvements for the HRA RECs that can be expanded to other regulatory contexts. These improvements might include:

1. Training on how to have a constructive discussion within committee, potentially facilitated by a structured Ethics Review Form (ERF) or similar.

2. Ensuring that the structure and order of documents submitted to RECs are consistent both in terms of the types of documents submitted for each study, and the internal consistency of all documents relating to a specific study.

3. A continued effort towards greater diversity among committee members, and where this is not possible a strengthened requirement for "Public and Patient Involvement and Engagement".

4. Regular feedback to committees on their performance.

## Supporting information

**S1 Data.**
(ZIP)

## Acknowledgments

The authors wish to thank the members of research ethics committees who contributed to the discussion, and support from Janet Messer, Jonathan Fennelly-Barnwell and Reshma Raycoba at the Health Research Authority.

### Human subjects research

The data collection described in this manuscript was performed as part of service improvement work by the Health Research Authority and therefore did not require an ethics review in accordance with the UK's Governance Arrangements for Research Ethics Committee's policy (section 2.3.13 & 2.3.14) [14].

## Author Contributions

**Conceptualization:** Mark Sidaway, Clive Collett, Simon Erik Kolstoe.

**Data curation:** Mark Sidaway.

**Formal analysis:** Mark Sidaway, Simon Erik Kolstoe.

**Project administration:** Mark Sidaway.

**Writing – original draft:** Simon Erik Kolstoe.

**Writing – review & editing:** Mark Sidaway, Clive Collett, Simon Erik Kolstoe.

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
