## [Decision Letter · Decision Letter 0]

27 Mar 2023

PONE-D-23-04580

Evidence from UK Research Ethics Committee members on what makes a good research ethics review, and what can be improved.

PLOS ONE

Dear Dr. Kolstoe,

Thank you for submitting your manuscript to PLOS ONE. After careful consideration, we feel that it has merit but does not fully meet PLOS ONE’s publication criteria as it currently stands. Therefore, we invite you to submit a revised version of the manuscript that addresses the points raised during the review process.

We have now received one review for your manuscript, but have been unable to secure a second.  So as not to delay the process any longer, as Academic Editor I have also undertaken a review. You will notice that I have focused on the introduction as I feel that the reviewer has clearly detailed the changes required to the other sections of the manuscript. 

Overall the manuscript needs significant revision to make it clear and understandable for a reader not an expert in the UK HRA and REC process. Related to this the discussion contains interpretation that appears to be based on expert personal reflections, rather than embedded within the relevant literature, and this needs addressing in the revisions. As highlighted by the reviewer the method section needs a major rewrite to include all the usual features of a methods section. 

We look forward to receiving your revised manuscript.

Kind regards,

Charlotte Lennox

Academic Editor

PLOS ONE

Journal Requirements:

    "The work described here was conducted as part of the normal duties of MS and CC as part of their employment with the Health Research Authority. SEK was seconded to the Health Research Authority to conduct this work, and was renumerated through the University of Portsmouth."

    "MS and CC work for the UK Health Research Authority. SEK was seconded to the HRA to conduct this research, and is also chair of the Cambridgeshire and Hertfordshire Research Ethics Committee. SEK is also a trustee of the charity UKRIO."

6. We note you have included a table to which you do not refer in the text of your manuscript. Please ensure that you refer to Table 1,4, 6 and 7in your text; if accepted, production will need this reference to link the reader to the Table.

Additional Editor Comments:

Introduction

Line 60 - RECS should be RECs

Line 67 - for a wider readership some more context is needed on 'Think Ethics', why was it established? What was the evidenced rationale for it? Is there a wider literature to support this, perhaps from other countries?

Line 70 - the four main findings need some additional detail as they lack clarity for example 'widespread support for alternative ethics review methods', how widespread, what alternative methods are being suggested - please add further details

Line 77 - If public contributors are not always familiar or experienced with/in the practicalities of conducting research ethics reviews, or indeed research itself - then the question is why was this public dialogue undertaken? It seems there is a level of assumed knowledge needed from the reader about the HRA.

Line 82 - ongoing projects seeking the opinion of REC members - such as? What's the rationale for focusing on this one in particular?

Line 83 -“REC members’ development days” - what are these?

Line 84 - These events were new to the HRA and their committees in 2022 - It is unclear which events are being referred to here. I am aware that training days for REC members existed even before the HRA existed, so would have assumed that a provision for REC members to come together existed under the HRA, prior to the pandemic - is this not the case? Perhaps some more detail of how the RECs operate and training etc is needed for the reader.

Line 91 - are the list of topics under discussion of relevance to the results? If not, you might want to remove this.

Reviewers' comments:

Reviewer's Responses to Questions

**Comments to the Author**

1. Is the manuscript technically sound, and do the data support the conclusions?

Reviewer #1: Partly

2. Has the statistical analysis been performed appropriately and rigorously? 

Reviewer #1: N/A

3. Have the authors made all data underlying the findings in their manuscript fully available?

Reviewer #1: Yes

4. Is the manuscript presented in an intelligible fashion and written in standard English?

Reviewer #1: Yes

5. Review Comments to the Author

Reviewer #1: This is an important topic area, and I am pleased to have been given the opportunity to review this manuscript.

I am in support of publication of this manuscript, subject to revision. At present the manuscript contains insufficient methodological information and the results section requires more synthesis and detail.

Please see my comments by section below.

Abstract

Line 39-37 It is unclear how ‘attending 37 annual training events’ is relevant here. Data were collected at 3 events.

Introduction

Line 49 State ‘Research Ethics Committees’ in full before using (REC).

Line 56 It would be helpful to define ‘research waste’.

Lines 60+ It would be useful here to provide an overview of the HRA review process. E.g., who are applicants, types of studies that require REC approval (as opposed to HRA only), panel members, role of the chair, use of ERF system etc. This will be common knowledge for most UK researchers who recruit via the NHS but not for those who don’t, or international readership, whose processes are likely to vary.

Material & Methods

Lines 100+ Most elements of a traditional methodology section are missing, namely headings and information related to: design, setting/participants, materials, procedure, analysis. Please reformat to ensure that all key elements of the methodology are present, clear, and replicable.

More information is required re. group facilitation and note taking. It is unclear why audio recording was not utilised so this should be considered as a limitation of the methodology.

The nature of the note taking needs for description. Exactly what was noted? Was this verbatim or summarised in the researchers’ words?

Line 125 A full description of the process of content analysis is required. I also note that this method of analysis is not supported by a reference. Which form of content analysis was used?

It is unclear how consensus themes were formed when analysis was conducted following two different procedures (NVivo/manual) – please provide more information. It is also unclear what manual means.

Results

Table 1. Best practice would be to collapse representation categories by region only, not by individual REC. For some RECs you have just 1 representative. Whilst it is unlikely that individuals’ can be identified, knowing that 1 member from a particular panel participants allows for process of elimination.

It is also unclear how these data were obtained as the methodology does not provide an overview of this data capture.

Line 134+ I note that much of the text here relates to respondents mentioning the procedures that underpin the ethical review process but not what your respondents perceived as facilitators/barriers to making ethical decisions. My assumption is that there is more context within the response data re. why the discussion is important, the importance of ‘consistency’, weighing up risk/benefit etc. I recommend revisiting the data and trying to provide a richer overview of the responses.

Line 136 Careful with use of terminology such as ‘theme’. We would expect this term to be referenced in a thematic analysis less so in a content analysis if this type.

Table 3. The theme names here provide little information without also containing a definition of the theme and/or a response example (this comment is relevant to all other theme tables.)

Line 157 ‘it was interesting’ – best practice is to avoid commentary/interpretation and state findings factually in the results section.

Line 162 The MHRA process will be unknown to most researchers. I think this highlights why it is important to provide an overview of the REC remit in the introduction, including its joint role with other agencies (e.g. MHRA, HMPPS etc.)

General comment:

As with my comment for Question 1, I imagine there is a lot more to unpack here other than respondents mentioning the same thing. What is it about these things that the respondents deemed important? E.g., ‘checking the clarity of the participant facing information was also a distinct theme’ – why? I also note that theme headings appear to overlap between Questions 1 and 2. I wonder if some further analysis is required here to move responses to the appropriate question? It is usual for respondents to provide comments to a question which are more relevant to a subsequent question.

It is accepted practice to re-group data where appropriate.

For example, I see ‘role of chair’ is a theme in Q2 but I imagine this response is more relevant to Q1.

Line 167-168 ‘a concern that committees need to stay focussed on fundamental issues, rather than being distracted by minor issues’ – this information is difficult to interpret if the reader does not understand the ethical review process (i.e. contentious ethical issues are discussed at panel and applicants receive comprehensive feedback on these and other minor issues in writing – this would be useful in the introduction).

Like 172 -174 This is presented as though it is a quotation, yet it is a researcher summary of responses, it should be clear that this is the case (this comment is relevant to all comment excerpts).

Lines 180-190 I see the ERF information is provided here which is great. I still recommend a simple description in the introduction, considering the ERF is mentioned in a previous question response.

201-202 What was/wasn’t seemed as useful about the headings?

Discussion

General comment:

There is much more detail provided re. the findings in the discussion than in the results section, where we should expect to see much of this summary (see comments about detail above). It would be useful to present the more of the discussion within the context of the literature base. My assumption is that the authors are very experienced REC members and much of the interpretation is personal reflection. It might be helpful to at least refer to REC terms of reference or policy documents when referring to process (assuming these are publicly accessible).

Line 329-333 Reference missing for these findings.

General comment:

Much of the discussion provided in the latter part of this section contains important information to aid HRA developments. However for a manuscript of this type I would expect to see application of these findings to policy for RECs in general. Are you able to apply these findings to other agencies E.g., university RECs, HMPPS, international agencies?

Line 374+ Please consider other limitations, e.g., the nature of the data collection.

6. PLOS authors have the option to publish the peer review history of their article (what does this mean?). If published, this will include your full peer review and any attached files.

Reviewer #1: No

---

## [Author Response · Author response to Decision Letter 0]

17 Apr 2023

17th April 2023

Dear Dr Lennox,

Thank you for arranging the review of our manuscript:

PONE-D-23-04580 Evidence from UK Research Ethics Committee members on what makes a good research ethics review, and what can be improved.

Peer review of a manuscript can sometime be a frustrating process, however in this case we were really appreciative of the constructive comments as they helped us to think carefully about this manuscript and what we are trying to present.

For some context, it is an interesting facet of research ethics review that committees are coordinated by administrative and/or regulatory authorities whose main aims is policy compliance and efficiency rather than academic transparency in the form of publications. As a consequence, despite the significant amount of work that goes into improving processes, the results are seldom released publicly as they instead take the form of updated standard operating procedures, guidance etc. that are seldom communicated in any meaningful way outside the organisation involved. This is a real weakness in the research system because organisations such as the UK’s Health Research Authority have considerable experience in research ethics review that is not well communicated.

To address this the aim of this paper was to take a really interesting piece of work conducted by the HRA as a service improvement activity, and try to make it more available for the wider research ethics community. The work was therefore not originally designed as a research project, and thus certain components – such as holding independent focus groups that could be recorded and transcribed – were not possible. However, despite these limitations, our aim was to turn what might have otherwise been a purely administrative (and potentially soon forgotten) project into a robust data collection and analysis that will hopefully meet the standard, and be of interest, to the literature and thus wider community.

The following describes our response to the review with reviewers comments in italic, and our comments marked in green (on submission we were asked to upload an unformatted version of this letter, but we note a version that retains the colours was included at the bottom of the pdf file).

Journal Requirements:

This has been done.

 "The work described here was conducted as part of the normal duties of MS and CC as part of their employment with the Health Research Authority. SEK was seconded to the Health Research Authority to conduct this work, and was remunerated through the University of Portsmouth."

The following is our revised funder statement:

“The work described here was conducted as normal duties for MS and CC as part of their employment with the Health Research Authority (HRA). SEK was seconded to the Health Research Authority to conduct this work, and was remunerated through the University of Portsmouth. As employees of the HRA this work was designed, and data collected, by the funder, however the analysis, decision to publish and preparation of the manuscript was conducted independently.”

 "MS and CC work for the UK Health Research Authority. SEK was seconded to the HRA to conduct this research, and is also chair of the Cambridgeshire and Hertfordshire Research Ethics Committee. SEK is also a trustee of the charity UKRIO."

We are happy to include this additional like in the Competing Interests section so that it now reads:

“MS and CC work for the UK Health Research Authority. SEK was seconded to the HRA to conduct this research, and is also chair of the Cambridgeshire and Hertfordshire Research Ethics Committee. SEK is also a trustee of the charity UKRIO. This does not alter our adherence to PLOS ONE policies on sharing data and materials.”

[text with various options deleted]

We have now made our data available as supplementary information.

This has been done.

6. We note you have included a table to which you do not refer in the text of your manuscript. Please ensure that you refer to Table 1,4, 6 and 7in your text; if accepted, production will need this reference to link the reader to the Table.

This has been done.

Additional Editor Comments:

Introduction

Line 60 - RECS should be RECs

Corrected

Line 67 - for a wider readership some more context is needed on 'Think Ethics', why was it established? What was the evidenced rationale for it? Is there a wider literature to support this, perhaps from other countries?

It has been clarified that the purpose of “Think Ethics” was to act on lessons learned from the COVID pandemic. Some references to other similar work have been included.

Line 70 - the four main findings need some additional detail as they lack clarity for example 'widespread support for alternative ethics review methods', how widespread, what alternative methods are being suggested - please add further details

These four findings were verbatim quotes from the report so have now been formatted as such to make this clear. Indeed the whole purpose of the work reported here was to try to determine what the “alternative review methods” might be from the perspective of REC members.

Line 77 - If public contributors are not always familiar or experienced with/in the practicalities of conducting research ethics reviews, or indeed research itself - then the question is why was this public dialogue undertaken? It seems there is a level of assumed knowledge needed from the reader about the HRA.

This has been further clarified to state that multiple stakeholder engagements have been conducted, one of which was the public involvement exercise referenced in the introduction, and another was the consultation with REC members which is being reported in this manuscript.

Line 82 - ongoing projects seeking the opinion of REC members - such as? What's the rationale for focusing on this one in particular?

This has been clarified as above – one project was seeking public opinion, this manuscript is focusing on the project seeking REC members opinions.

Line 83 -“REC members’ development days” - what are these?

Line 84 - These events were new to the HRA and their committees in 2022 - It is unclear which events are being referred to here. I am aware that training days for REC members existed even before the HRA existed, so would have assumed that a provision for REC members to come together existed under the HRA, prior to the pandemic - is this not the case? Perhaps some more detail of how the RECs operate and training etc is needed for the reader.

Line 91 - are the list of topics under discussion of relevance to the results? If not, you might want to remove this.

The description of the REC members days has been expanded, although the final comment (regarding old Line 91) is slightly inconsistent with the earlier request to describe what these days entail – but hopefully this section now reads a bit better.

5. Review Comments to the Author

Abstract

Line 39-37 It is unclear how ‘attending annual training events’ is relevant here. Data were collected at 3 events.

This has been clarified.

Introduction

Line 49 State ‘Research Ethics Committees’ in full before using (REC).

Line 56 It would be helpful to define ‘research waste’.

Both now done.

Lines 60+ It would be useful here to provide an overview of the HRA review process. E.g., who are applicants, types of studies that require REC approval (as opposed to HRA only), panel members, role of the chair, use of ERF system etc. This will be common knowledge for most UK researchers who recruit via the NHS but not for those who don’t, or international readership, whose processes are likely to vary.

This description has been significantly expanded as requested.

Material & Methods

Lines 100+ Most elements of a traditional methodology section are missing, namely headings and information related to: design, setting/participants, materials, procedure, analysis. Please reformat to ensure that all key elements of the methodology are present, clear, and replicable.

Headings have been added, and care taken to ensure that the description is sufficient to allow another research team to replicate the activities described here.

More information is required re. group facilitation and note taking. It is unclear why audio recording was not utilised so this should be considered as a limitation of the methodology.

The nature of the note taking needs for description. Exactly what was noted? Was this verbatim or summarised in the researchers’ words?

The method used has been expanded and the consequent limitations described more fully. However, as noted in the introductory paragraph of this letter, this manuscript is reporting on an activity originally designed for service improvement and not research per se, and thus while we are confident that the results are robust, we acknowledge that they do not follow standard research practice.

Line 125 A full description of the process of content analysis is required. I also note that this method of analysis is not supported by a reference. Which form of content analysis was used?

It is unclear how consensus themes were formed when analysis was conducted following two different procedures (NVivo/manual) – please provide more information. It is also unclear what manual means.

Thank you for this comment. You are absolutely correct that some confusion crept in by us referring to “themes”, so we have now significantly tidied up the results and discussion section to make sure it is clear that we conducted a content analysis and thus inductively identified “categories” of comments, rather than classified comments into pre-defined themes. We have referenced both a classic paper on the complexities of this approach alongside a very recent paper in the journal Research Ethics that conducted a similar analysis to the one we have done here. Regarding “manual coding” this was more a passing (and perhaps overly honest) reflection of trouble that one author had operating NVivo, meaning that the other researcher had to manually load all the categories into the software. Reference to this has now been deleted as despite taking a long time, it did not have any meaningful impact on the results.

Results

Table 1. Best practice would be to collapse representation categories by region only, not by individual REC. For some RECs you have just 1 representative. Whilst it is unlikely that individuals’ can be identified, knowing that 1 member from a particular panel participants allows for process of elimination.

It is also unclear how these data were obtained as the methodology does not provide an overview of this data capture.

Both done – table categories collapsed, and a line added to the methods describing how this data was obtained.

Line 134+ I note that much of the text here relates to respondents mentioning the procedures that underpin the ethical review process but not what your respondents perceived as facilitators/barriers to making ethical decisions. My assumption is that there is more context within the response data re. why the discussion is important, the importance of ‘consistency’, weighing up risk/benefit etc. I recommend revisiting the data and trying to provide a richer overview of the responses.

I think this reflects the challenges of this type of data collection and analysis. A significant weakness of our method (now explicitly stated and discussed) was the use of note takers who often did not provide the full context of specific comments. We have therefore tried to draw out as much as this as we can, and added a few more direct quotes from the data, but are mindful of other feedback that you have given about not using our own experience to overly interpret the data. However, should the reader be interested in trying to glean further context, the data has now been included in supplementary materials.

Line 136 Careful with use of terminology such as ‘theme’. We would expect this term to be referenced in a thematic analysis less so in a content analysis if this type.

Table 3. The theme names here provide little information without also containing a definition of the theme and/or a response example (this comment is relevant to all other theme tables.)

Have replaced all use of the word “theme” with “category” to reflect that this is a content, not thematic, analysis.

Line 157 ‘it was interesting’ – best practice is to avoid commentary/interpretation and state findings factually in the results section.

Deleted in the results, although kept a couple instances of the word “interesting” in the discussion where we feel it is more acceptable to provide interpretation.

Line 162 The MHRA process will be unknown to most researchers. I think this highlights why it is important to provide an overview of the REC remit in the introduction, including its joint role with other agencies (e.g. MHRA, HMPPS etc.)

Done

General comment:

As with my comment for Question 1, I imagine there is a lot more to unpack here other than respondents mentioning the same thing. What is it about these things that the respondents deemed important? E.g., ‘checking the clarity of the participant facing information was also a distinct theme’ – why? I also note that theme headings appear to overlap between Questions 1 and 2. I wonder if some further analysis is required here to move responses to the appropriate question? It is usual for respondents to provide comments to a question which are more relevant to a subsequent question.

It is accepted practice to re-group data where appropriate.

For example, I see ‘role of chair’ is a theme in Q2 but I imagine this response is more relevant to Q1.

We have provided some comment on this, although as mentioned, because of the limitation of using note takers we have been quite strict with ourselves in trying not to over interpret the data. As a consequence we have kept the results and initial discussion focussed purely on the data that actually came in under each heading, although our overall conclusions does take a slightly broader view of ideas that were expressed across the five questions (see point below regarding the four “actions” that the HRA will be taking away from this work).

Line 167-168 ‘a concern that committees need to stay focussed on fundamental issues, rather than being distracted by minor issues’ – this information is difficult to interpret if the reader does not understand the ethical review process (i.e. contentious ethical issues are discussed at panel and applicants receive comprehensive feedback on these and other minor issues in writing – this would be useful in the introduction).

Like 172 -174 This is presented as though it is a quotation, yet it is a researcher summary of responses, it should be clear that this is the case (this comment is relevant to all comment excerpts).

Lines 180-190 I see the ERF information is provided here which is great. I still recommend a simple description in the introduction, considering the ERF is mentioned in a previous question response.

201-202 What was/wasn’t seemed as useful about the headings?

All four points clarified in the text.

Discussion

General comment:

There is much more detail provided re. the findings in the discussion than in the results section, where we should expect to see much of this summary (see comments about detail above). It would be useful to present the more of the discussion within the context of the literature base. My assumption is that the authors are very experienced REC members and much of the interpretation is personal reflection. It might be helpful to at least refer to REC terms of reference or policy documents when referring to process (assuming these are publicly accessible).

We have tried to make the discussion a bit more factual and added some more references. However, it should be noted that as a service improvement exercise, the outcome as far as the HRA are concerned will/has been succinctly summarised in their internal report (not published externally) as being:

1) Provide more training on how to have a constructive discussion within the committee, updating the Ethics Review Form (ERF) to help guide such discussion.

2) Review the structure of study documents to ensure the material given to committees is more consistent

3) Continue working towards greater diversity among committee members, and where this is not possible strengthen the requirement for “Public and Patient Involvement and Engagement”.

4) Provide more feedback to committees on their performance.

We have therefore structured the discussion around these four points, showing how the data has led to these recommendations. Indeed as these conclusions are generalisable well beyond the HRA RECs, one big motivation for submitting this manuscript was to highlight the evidence for these so that they could be used elsewhere. We are not aware that this particular journal allows summary boxes, but would be happy to include these four recommendations in a more obvious way if the reviewer feels they are not displayed suitably. They are, however, stated in the abstract.

Line 329-333 Reference missing for these findings.

Reference added.

General comment:

Much of the discussion provided in the latter part of this section contains important information to aid HRA developments. However for a manuscript of this type I would expect to see application of these findings to policy for RECs in general. Are you able to apply these findings to other agencies E.g., university RECs, HMPPS, international agencies?

See above comment – we do think these findings are broadly generalisable across other RECs (and IRBs), and hence the reason for this paper.

Line 374+ Please consider other limitations, e.g., the nature of the data collection.

Now added as described above.

(final comment from editor):

This has been done and the modified figure file from the PACE tool uploaded.

We once again thank the reviewers for their comments that we do feel have genuinely improved our manuscript, and hope that it is now in a position to be published.

Yours Sincerely,

Simon, Matt and Clive.

---

## [Decision Letter · Decision Letter 1]

15 May 2023

PONE-D-23-04580R1Evidence from UK Research Ethics Committee members on what makes a good research ethics review, and what can be improved.PLOS ONE

Dear Dr. Kolstoe,

Thank you for submitting your manuscript to PLOS ONE. After careful consideration, we feel that it has merit but does not fully meet PLOS ONE’s publication criteria as it currently stands. Therefore, we invite you to submit a revised version of the manuscript that addresses the points raised during the review process. We'd like to thank you for addressing the first round of comments in such detail.  The manuscript has been reviewed by the same reviewer(s) and they have identified some additional minor comments. As Academic Editor I do agree that these changes would benefit the manuscript, in terms of boosting the scientific rigor of the method and results.   

We look forward to receiving your revised manuscript.

Kind regards,

Charlotte Lennox

Academic Editor

PLOS ONE

Journal Requirements:

Reviewers' comments:

Reviewer's Responses to Questions

**Comments to the Author**

1. If the authors have adequately addressed your comments raised in a previous round of review and you feel that this manuscript is now acceptable for publication, you may indicate that here to bypass the “Comments to the Author” section, enter your conflict of interest statement in the “Confidential to Editor” section, and submit your "Accept" recommendation.

Reviewer #1: (No Response)

2. Is the manuscript technically sound, and do the data support the conclusions?

Reviewer #1: Yes

3. Has the statistical analysis been performed appropriately and rigorously? 

Reviewer #1: N/A

4. Have the authors made all data underlying the findings in their manuscript fully available?

Reviewer #1: No

5. Is the manuscript presented in an intelligible fashion and written in standard English?

Reviewer #1: Yes

6. Review Comments to the Author

Reviewer #1: I am pleased to see that the authors have gone to great effort to address my comments and make changes to the manuscript.

I have a few further comments below which can be addressed via minor revision.

Materials and methods:

I can see great improvement here. Whilst I recognise that this data collection was not intended as research activity, there are still some details missing.

• Setting/participants – participants are not described. Who attended? How many?

• Design – the activity design remains unnamed. Perhaps ‘consultation exercise’ in appropriate.

• Procedure – more details are required re. note taking procedure.

• Analysis – I note you reference Braun and Clark (15) for content analysis however the paper referenced concerns thematic analysis. Please provide an appropriate reference.

• Analysis – more detail is required re. the content analysis procedure. Each step of grouping of content should be described.

• Analysis – this section, or the beginning of the results section should state how the findings are presented and explicitly state that the quotes included are note excerpts and not verbatim participant quotes.

Results:

Line 193-195: ‘although which aspects of risk and safety were most important was difficult to assess as notes often just stated “risk” or similar one-word answers’ – I recommend omitting this sentence and instead discussing the quality of note taking as a limitation in the discussion.

Discussion:

Line 384+:

‘It is unclear whether RECs themselves should be entirely representative of society due to the technical nature of documents under review, and also the specific skills needed by members to operate within the committee review structure’.

- I understand what you are getting at here, but I feel there is opportunity for misinterpretation. A less considered reader might surmise that you think groups with particular characteristics are less likely to have the skills required.

it might be pertinent here to reflect on professional roles of REC chair and members (clinical, academic etc.) and how these professionals are not representative of society, therefore it is perhaps unsurprising that the REC panel is not representative – then move onto praising the inclusion of lay and PPIE members.

Line 223 – a conclusion paragraph would be of great value here – with recommendations for future REC practice/HRA oversite/research into the area.

7. PLOS authors have the option to publish the peer review history of their article (what does this mean?). If published, this will include your full peer review and any attached files.

Reviewer #1: No

---

## [Author Response · Author response to Decision Letter 1]

23 May 2023

(Please note: a version of this letter with formatting can be found in the pdf version of this submission as it is difficult to distinguish between the reviewers requests and our responses without formatting)

Dear Dr Lennox,

Thank you for the feedback requesting minor revisions of our manuscript:

PONE-D-23-04580 Evidence from UK Research Ethics Committee members on what makes a good research ethics review, and what can be improved.

We believe we have now addressed the final minor revisions as follows:

Journal Requirements:

These have been checked and updated.

Reviewer #1: I am pleased to see that the authors have gone to great effort to address my comments and make changes to the manuscript.

I have a few further comments below which can be addressed via minor revision.

Thank you for your reviews and further comments that are definitely contributing to the quality of this manuscript. Much like a PhD viva it can be quite fun engaging with someone interested in our research!

Materials and methods:

I can see great improvement here. Whilst I recognise that this data collection was not intended as research activity, there are still some details missing.

• Setting/participants – participants are not described. Who attended? How many?

This is now explicitly stated.

• Design – the activity design remains unnamed. Perhaps ‘consultation exercise’ in appropriate.

• Procedure – more details are required re. note taking procedure.

Both done.

• Analysis – I note you reference Braun and Clark (15) for content analysis however the paper referenced concerns thematic analysis. Please provide an appropriate reference.

This comment took us down a rabbits warren trying to define exactly what our specific analysis should be called. We now reference two updated papers by Braun and Clark (along with their much referenced 2006 paper), where in their most recent discussion published in January this year they state: 

We suggest that a relativist approach to quality facilitates rigour, through requiring a thoughtful and knowing researcher, who engages and reflects, considering quality in the context of a particular study, rather than having a checklist of universal standards to meet. (https://doi.org/10.1080/17437199.2022.2161594). 

As a consequence we have focussed on whether, within the context of our work, our analysis is robust, repeatable and the conclusions are in actual fact relevant, useful and helpful to the development of REC review. There will of course always be other ways of conducting the analysis, but we believe our approach has been suitable. We are therefore thankful for your two final suggestions relating to the analysis that we have also addressed, seeking to reinforce the description and therefore reproducibility of our approach:

• Analysis – more detail is required re. the content analysis procedure. Each step of grouping of content should be described.

• Analysis – this section, or the beginning of the results section should state how the findings are presented and explicitly state that the quotes included are note excerpts and not verbatim participant quotes.

Moving on to the results section:

Line 193-195: ‘although which aspects of risk and safety were most important was difficult to assess as notes often just stated “risk” or similar one-word answers’ – I recommend omitting this sentence and instead discussing the quality of note taking as a limitation in the discussion.

Done – second half of sentence deleted.

Discussion:

Line 384+: ‘It is unclear whether RECs themselves should be entirely representative of society due to the technical nature of documents under review, and also the specific skills needed by members to operate within the committee review structure’. - I understand what you are getting at here, but I feel there is opportunity for misinterpretation. A less considered reader might surmise that you think groups with particular characteristics are less likely to have the skills required. it might be pertinent here to reflect on professional roles of REC chair and members (clinical, academic etc.) and how these professionals are not representative of society, therefore it is perhaps unsurprising that the REC panel is not representative – then move onto praising the inclusion of lay and PPIE members.

Done – sentence deleted and the roles of expert (professional)/ lay REC members emphasised.

Line 223 – a conclusion paragraph would be of great value here – with recommendations for future REC practice/HRA oversite/research into the area.

Done – conclusions/recommendations added as in our previous letter to you.

We thank you again for these final updates to our manuscript, and hope that with these minor revisions addressed we will soon be able to see the paper published.

Yours Sincerely,

Simon, Matt and Clive.

---

## [Editor Report · Decision Letter 2]

19 Jun 2023

Evidence from UK Research Ethics Committee members on what makes a good research ethics review, and what can be improved.

PONE-D-23-04580R2

Dear Dr. Kolstoe,

We’re pleased to inform you that your manuscript has been judged scientifically suitable for publication and will be formally accepted for publication once it meets all outstanding technical requirements.

Kind regards,

Simon White

Academic Editor

PLOS ONE
---

## [Editor Report · Acceptance letter]

23 Jun 2023

PONE-D-23-04580R2 

Evidence from UK Research Ethics Committee members on what makes a good research ethics review, and what can be improved. 

Dear Dr. Kolstoe:

I'm pleased to inform you that your manuscript has been deemed suitable for publication in PLOS ONE. Congratulations! Your manuscript is now with our production department. 

Kind regards, 

on behalf of

Dr. Simon White 

Academic Editor

PLOS ONE